# Effects of Physical Activity on Brain Energy Biomarkers in Alzheimer’s Diseases

**DOI:** 10.3390/diseases8020018

**Published:** 2020-06-08

**Authors:** Khadijeh Ebrahimi, Morteza Jourkesh, Saeed Sadigh-Eteghad, Stephen R Stannard, Conrad P. Earnest, Roger Ramsbottom, Jose Antonio, Khan H. Navin

**Affiliations:** 1Department of Sports Science and Physical Education, Marand Branch, Islamic Azad University, Marand 5418916571, Iran; kh.ebrahimi90@gmail.com; 2Department of Physical Education & Sports Science, Islamic Azad University, Shabestar Branch 5381637181, Iran; 3Neurosciences Research Center, Tabriz University of Medical Sciences, Tabriz 5166614766, Iran; s.sadigh@tbzm.ac.ir; 4School of Sport, Exercise, and Nutrition, Massey University, Palmerston North 4442, New Zealand; srstannard@gmail.com; 5Department of Health and Kinesiology, Texas A&M University, College Station, TX 77843, USA; cpe.sci@icloud.com; 6Department of Sport, Health Sciences and Social Work, Oxford Brookes University, Oxford OX3 0BP, UK; rramsbottom@brookes.ac.uk; 7Department of Health and Human Performance, Nova Southeastern University, Davie, FL 32004, USA; ja839@nova.edu; 8Immune Whey, Sugar Hill, GA 30518, USA; nkhan30518@gmial.com

**Keywords:** Alzheimer’s, physical activity, prevention and memory

## Abstract

The prevalence of dementia has substantially increased worldwide. Currently, there is no cure for dementia or Alzheimer’s disease (AD), and care for affected patients is financially and psychologically costly. Of late, more attention has been given to preventive interventions—in particular, physical activity/exercise. In this review, examine the risk factors associated with AD and the effects physical activity may play in the prevention of the degenerative process of this disease, loss of memory and cognitive performance in the elderly. To date, research has shown that physical activity, especially aerobic exercise, has a protective effect on cognitive function and memory in the elderly and Alzheimer’s patients. In comparison with aerobic exercise, several strength training studies have also shown positive effects, and the rare studies that compare the two different modalities show no difference.

## 1. Introduction

Alzheimer’s disease (AD) is the most common cause of dementia and is an age-related neurodegenerative disease which leads to the loss of memory and learning in mid-to-late life [1].Globally, over 50 million people have dementia, Alzheimer’s disease being the most common form and responsible for 60–70% of cases (WHO) [2]. Currently, about ten percent of the population over 65 years, more than 25 million people worldwide, are affected by this disease. More than a century has passed since specialist neurologist Alois Alzheimer described, for the first time, a 51-year-old female patient with a progressively sharp memory decline, brain atrophy, senile plaques (SPs, or neuritic plaque), and neurofibrillary tangles (NFTs) [3]. Senile plaques and NTFs consist of aberrant cumulative extracellular amyloid β-protein (Aβ) and the hyper-phosphorylation of intracellular microtubule-associated protein tau [4]. The most affected areas in the brains of AD patients include the olfactory bulb, neocortex, and hippocampus, [3] which plays a leading role in spatial memory and learning [5]. Despite intensive research into AD and other neurological diseases, no drug has yet been developed to effectively treat all the pathological aspects of the disease or reduce the risk of age-related cognitive disorders and dementia [6]. According to the literature, factors that protect the heart also protect the brain and this may reduce the risk of AD and other dementia-related diseases [7]. Physical activity, especially aerobic exercise, is one such factor [7] High- and moderate-intensity exercise training also improves cerebrovascular reactivity and this is important for memory, executive function, and mental health [8]. Therefore, the purpose of this article is to review AD risk factors and detail the effect of physical activity on learning, cognitive function and molecular factors affecting Alzheimer’s disease.

## 2. Risk Factors for Alzheimer’s Disease 

Alzheimer’s disease is a complex disorder, which may be due to a combination of genetic, biochemical and environmental factors. The biggest risk factor for AD, age, is unmodifiable but the disease is not a normal part of the aging process. Many Alzheimer’s patients have no family history of the disease, yet are diagnosed with AD at approximately the age of 65 years and classified as having sporadic Alzheimer’s disease (SAD) [7,9]. 

Without the benefit of experimental data, useful observational studies rely on large numbers of study participants, and meta-analyses to give weight to the evidence. In this regard, the publication of Xu et al. (2015) provides some clues as to modifiable risk factors in the development of AD. The inclusion of 232 separate studies provide good statistical power, though the results appear inconsistent. For example, whilst physical activity was seen to be protective, low body mass index (BMI) was a risk factor, yet physical activity and BMI levels have been shown to be inversely related to the development of AD [10] Surprisingly, the Xu et al. (2015) meta-analysis showed heart disease and cancer to be be ‘protective’, but possibly because mortality associated with these reduce the age-related development of AD, or because identification and treatment of these diseases was somehow protective against AD. Nevertheless, and in accord with previous evidence [11], cognitive activity, intake of some specific nutrients (e.g., folate, fish), and physical activity were considered protective. Meanwhile, the impact of light smoking in the development of AD is less clear, although heavy smoking is a common and major overall risk factor [12].

Some specific genetic factors appear to be involved in the occurrence of AD, including the gene apolipoprotein E (APOE) [13,14]. APOE has three major alleles: ε2, ε3 and ε4 and it has been reported that expression of the ε4 allele leads to an accelerated loss of nerve function and decreases the onset age of AD [15,16]. About one percent of AD cases are inherent and thus classified as familial AD (FAD). FAD is associated with mutations of the genes encoding amyloid-β precursor protein (APP), presenilin 1 (PS1) or presenilin 2 [9]. 

## 3. Aetology of Alzheimer’s Disease

The most important theory proposed to explain the development of AD is the "amyloid cascade hypothesis" (ACH), which was suggested for the first time in 1992 [17]. According to the ACH theory, the primary pathological event in AD includes A*β* peptide production and deposition by APP in the brain parenchyma and cerebrospinal fluid, which leads to the formation of SPs, then NFTs, the death of neurons and, eventually, dementia [18,19]. There are a wide variety of A*β* forms, including A*β*40 with 40 amino acids and the peptide A*β*42, the most common type, which is significantly more toxic than A*β*40 [20]. On the other hand, A*β*42 is the major variant of A*β* in the core of senile plaques, whereas A*β*40 constitutes ∼90% of total A*β* in plasma in which, in most FAD cases, the ratio of A*β*42/A*β*40 is shown to be elevated [21].

APP is cleaved by two alternative pathways: an amyloidogenic and a non-amyloidogenic pathway. In the amyloidogenic pathway, APP is sequentially cleaved by a *β*-secretase (BACE1) to generate two cleavage fragments, APPs*β* and C99 [22,23]. Successive cleavage of C99 is undertaken by γ-secretase, generating neurotoxic A*β* peptides [3,22]. Increased A*β* production contributes to the etiological basis of Alzheimer’s disease [24]. APP cleavage by α-secretase (via the non-amyloidogenic pathway) generates two fragments, APPsα and C83. C83 is subsequently cleaved by γ-secretase, generating the pathologically irrelevant p3 peptide [25]. The presenilins (PS1 and PS2) are the catalytic core of the γ-secretase, so presenilin mutations—seen in Alzheimer’s disease—could alter the activity of γ-secretase and Aβ production [26]. Moreover, presenilin mutations could result in increasing the ratio of Aβ42/Aβ40 [9]. 

Over the last five to six years, alternatives to the ACH hypothesis have emerged, namely a theory suggested by Morris and coworkers (2014) providing evidence that the process of (neuro)inflammation may be a “major player” in the pathogenesis of AD, independent of A*β* formation [27] . In a similar vein, Clark et al. (2018) suggested a growth area for research into the treatment of neurodegenerative disease (e.g., post-stroke syndrome, traumatic brain injury (TBI), AD), could be to reduce chronically elevated levels of Tumor Necrosis Factor (TNF), and other cytokines—elevated levels of which, in turn, generate A*β* [28] . While Morris et al. (2014, 2018) acknowledge that A*β* status may indeed predict the risk of dementia, they go on to suggest that a greater holistic approach to cognitive decline could further our understanding of this disease [27,29]. 

In addition to the aforementioned risk factors, obesity, cardiovascular disease, elevated cholesterol, hypertension and sedentary lifestyles also play a role in the development of AD [30,31,32].

Of those, a lack of physical activity is an important risk factor for the onset of dementia and Alzheimer’s disease [32,33]. A summary of AD risk and protective factors is shown in Figure 1.

## 4. Impaired Brain Creatine Kinase Activity and Cerebral Glucose Metabolism

Lowered levels of brain creatine kinase activity and reduced cerebral PCr levels are associated with AD [35] and could be considered a risk factor in diagnosis. Creatine kinase (CK) activity is vital for the energy reaction of every cell in the human body as a spatial energy shuttle and energy sensor [35] and is thus paramount in bioenergetics of the brain [35]. 

The cerebral form of CK is known as cytosolic brain-type creatine kinase (BB-CK). AD patients manifest high levels of cytosolic brain-type creatine kinase (BB-CK) oxidative damage [35,36]. BB-CK activity in AD patients is reduced by to up to 86%, coinciding with a 14% reduction in CK Protein expression [36] Reduced BB-CK activity in AD results in decreased ATP stores in neural cells and synapses [35,36] BB-CK and the simultaneous expression of ubiquitous mitochondrial creatine kinase activity (uMtCK) is reduced in AD patients via high levels of oxidation, manifesting in BB-CK enzyme dysfunction and the progression of AD [35]. AD patients have mitochondrial DNA mutations manifesting in the dysfunctional cerebral bioenergetics of the brain [35] in essence, BB-CK activity is paramount in neuronal energetics, the facilitation of synaptic glutamate uptake and, ultimately, neurotransmitter uptake [35]. 

Another cerebral bioenergetic risk factor for AD is reduced cerebral glucose metabolism and mitochondrial DNA mutation [35,36].AD-associated mitochondrial DNA mutations result in perturbed energy metabolism in the brain and impaired central nervous system (CNS) function. Reduced acetyl-CoA production, cortical acetylcholine esterase activity, and oxidative phosphorylation are important risk factors in the onset of AD-associated senile dementia [35].

## 5. The Effect of Physical Activity on Alzheimer’s Disease

Evidence from both human and animal studies suggests that physical activity, especially exercise that increases cardiorespiratory fitness, facilitates the neuroplasticity of certain brain structures related to cognitive function [37]. In a small, randomized controlled trial (RCT) study, six months of high-intensity aerobic exercise (75-85% heart rate reserve) in patients with mild cognitive impairment (MCI), decreased the plasma concentration of A*β*-42 [38]. Furthermore, A*β*-dependent neuronal cell death in the hippocampus of NSE/PS2m mice (Tg) mice was markedly suppressed following treadmill exercise for 12 weeks from 24 months of age. These data strongly suggest that exercise provides a therapeutic potential for inhibiting both A*β*-42 and pathways of neuronal death [39]. 

In rodent models, swim training five days/week, 1 h/day, for six weeks decreased tau phosphorylation and APP expression and improved spatial learning and memory in diabetic rats [40]. Similarly, three weeks of voluntary wheel running significantly decreased soluble A*β*40 and soluble fibrillar A*β* in aged Tg2576 mice (17–19 months), and the authors concluded that treadmill exercise may be beneficial in the prevention or treatment of AD [41], although experimental evidence is still needed from corresponding human studies. Consistently, A*β*-42 peptides decreased significantly in the NSE/APPsw Tg mice following exercise on a treadmill for 16 weeks. Furthermore, ten weeks of treadmill training in 1.5- to 4-month-old APP/PS1 transgenic (Tg) mice is known to enhance hippocampus-associated memory and amygdala-associated neuronal function and reduce the levels of soluble A*β* in the amygdala and hippocampus and serves as a means to delay the onset of AD [1].

It has also been demonstrated that treadmill exercise (TE) prevented PS2 mutation-induced memory impairment and reduced A*β*-42 deposition through the inhibition of *β*-secretase (BACE-1) and its product, C-99 in the cortex and/or hippocampus of aged PS2 mutant mice [42]. In addition, five months of treadmill exercise resulted in a robust reduction in *β*-amyloid (A*β*) deposition and tau phosphorylation in the hippocampus of APP/PS1 mice, which was also accompanied by a significant decrease in APP phosphorylation and PS1 expression. Thus, long-term treadmill exercise seems to mediate APP processing in favor of reduced A*β* deposition in animal models [43]. 

In human studies, it has been shown that there is a novel interaction between APOE status and exercise engagement [44,45]. Therefore, regular physical activity may reduce the risk or delay the onset of dementia and AD, especially among APOE ε4 allele carriers [46]. It has been suggested that exercise prevents the decline of neurovascular structure with age, but not in the absence of APOE. In other words, exercise has little or no effect on these changes in the absence of APOE [47]. A recent narrative review suggests exercise training and physical activity may have a significant role in the prevention of AD [48]. Taken together, these results suggest that exercise training represents a practical therapeutic strategy for humans suffering from AD [33], although differing modalities need to be explored with respect to effectiveness.

## 6. Exercise, Memory and Learning 

There are many studies examining the effect of physical activity on memory and learning using rodent models. For example, five months of voluntary wheel running decreases extracellular amyloid-*β* (A*β*) plaques in the frontal cortex and enhances the rate of learning and memory in TgCRND8 animals negotiating the Morris water maze, with significant reductions in escape latencies over the first three (of six) trial days [49]. Voluntary wheel running for ten weeks also reduced all the neuropathological hallmarks of AD, reduced neuronal loss, increased hippocampal neurogenesis and reduced spatial memory loss in a double-transgenic APPswe/PS1ΔE9 mouse model of AD [50]. Lastly, four weeks of treadmill exercise prevented learning and memory impairment and the suppression of early long-term potentiation of CA1 area pyramidal cells in Alzheimer’s disease-like pathology, again demonstrated in a rodent model [51]. 

In the aged animal, exercise is a very useful strategy for preventing memory failure. In one study, treadmill exercise improved short-term and spatial memories by enhancing neurogenesis and suppressing apoptosis in the hippocampal dentate gyrus of old-aged rats [52]. Consistently short bouts (4–6 min) of mild-intensity physical exercise during five consecutive weeks imporved spatial learning and memory in ageing rats [53]. Moreover, Van Praag et al. (2005) showed that voluntary exercise ameliorates some of the deleterious morphological and behavioral consequences of ageing in which a decline in memory in aged mice was reversed by running [54]. Therefore, aerobic exercise training, e.g., running, may improve memory and learning and be beneficial in reducing the risk or delaying the onset of dementia and AD in mice [39] and humans [39,46]. Another study carried out recently on rats suggested that voluntary resistance wheel running (to a maximum load of 30% of body mass) for four weeks led to improved spatial learning and memory and thus plays a beneficial role in hippocampus-related cognitive functions [55].

In contrast relatively few human intervention studies have examined resistance training with regard to learning and memory, dementia and AD. While resistance training has been performed in older groups, it has not been undertaken in individuals presenting with AD. For example, it has been shown that twelve months of resistance training in older women (65–75 years) may be a promising candidate for preventing cognitive decline and increasing cognitive performance [56,57]. Earlier results in elderly humans reported by Cassilhas et al. (2007) showed a significant and positive impact of resistance training at two different intensities (moderate and high) on cognitive function, as well as improving physical function. However, moderate-intensity exercise was more effective compared with high-intensity exercise for improving mood profile in the elderly [58]. A study by Perrig-Chiello et al. (1998) also resulted in improved cognitive function following an eight week resistance exercise program in 46 elderly volunteers (average age 73.2 years) [59]. 

A comparison between resistance and aerobic training has shown that both modes have a positive effect on cognitive function, as demonstrated in 36 volunteers aged 60–85 years performing 9 weeks of physical exercise. There was no difference between the two exercise groups (resistance versus endurance), suggesting that engaging in either form of exercise is beneficial, which may influence patient adherence based on personal preference [60]. Similar results have also been observed in a rodent model (90-day-old rats) undertaking eight weeks of aerobic training on a treadmill and resistance exercise on a vertical ladder, showing an improvement in learning and spatial memory in a similar manner, using either the Morris water maze test or a passive avoidance task [61]. A simplified diagram of how exercise may simultaneously improve memory, higher executive functioning, and act to reduce cognitive impairment, acting via multiple pathways to reduce inflammation [28] and enhance neural plasticity, is suggested by Figure 2.

## 7. Conclusions

There is strong evidence to suggest exercise (in the form of endurance and strength training) appears to be a helpful non-pharmacological approach that delays dementia and Alzheimer’s disease. The results of studies (both animal and human) in recent years show a positive effect of exercise on cognitive function and memory, as well as reducing risk factors for dementia and AD. However, there is limited research into the effects of resistance training on memory, learning and specific risk factors for Alzheimer’s disease—the main finding of the existing (limited) research was that strength training had a positive effect on cognitive function and memory. Finally, those studies comparing the two modalities (endurance vs. resistance) found no difference; however, there is a paucity of studies to this effect and future research efforts should further consider these two modalities in the field of memory and learning, especially in AD. Ultimately, it is not known which type of exercise (strength or endurance) is more effective in this area and which could take precedence.

## Figures and Tables

**Figure 1 diseases-08-00018-f001:**
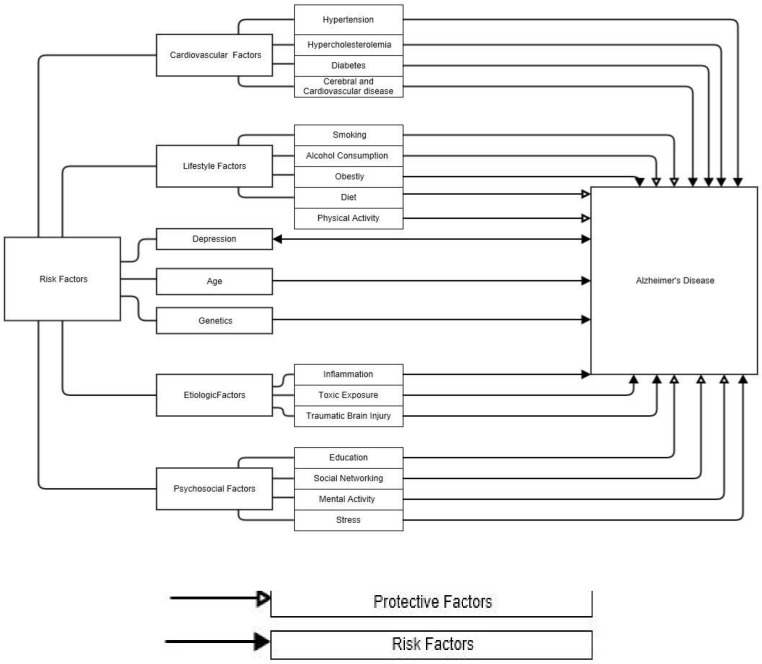
Protective and risk factors associated with Alzheimer’s disease [34].

**Figure 2 diseases-08-00018-f002:**
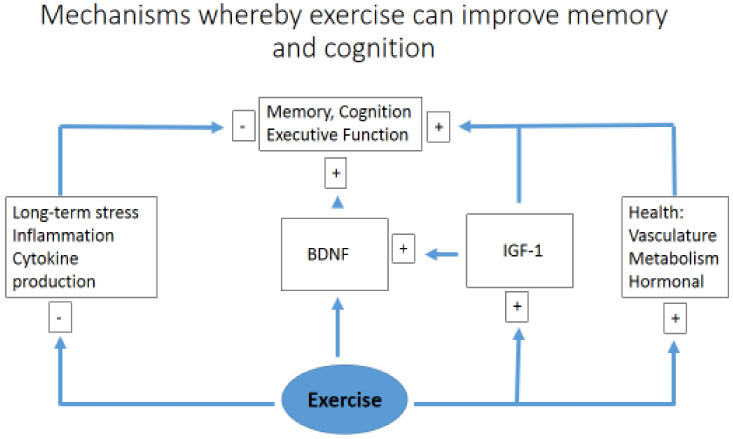
A simplified diagram of how exercise may simultaneously improve memory and cognition. Brain-derived neurotrophic factor (BDNF); insulin-like growth factor 1 (IGF1).

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
