# Peer review of "Effects of Physical Activity on Brain Energy Biomarkers in Alzheimer’s Diseases"

_diseases, 2020, doi:10.3390/diseases8020018_

Round 1

Reviewer 1 Report

In this review, the authors examined the risk factors associated with Alzheimer’s disease (AD) and the effects physical activity in the prevention of this neurodegenerative process. The paper is well organized and written. As a suggestion to improve the draft, please add a paragraph examining the beneficial effects of physical exercise on AD-related proteins in human subjects and in particular in blood cells, taken as peripheral model (see for example Exerc Immunol Rev. 2019;25:34-49. Front Aging Neurosci. 2018 Jan 30;10:17; Mol Neurobiol. 2018 Mar;55(3):2653-2675; Exp Gerontol. 2018 Jul 15;108:62-68). Of particular interest, papers examining the above-mentioned effects in people with different ApoE polymorphisms (Antioxidants (Basel). 2019 Nov 9;8(11). pii: E538; BMC Genomics. 2017 Nov 14;18(Suppl 8):803; Front Endocrinol (Lausanne). 2018 Mar 6;9:71. doi: 10.3389/fendo.2018.00071).

Author Response

Please see the responses as attach file.

All the best,

Dr.Morteza jourkesh

Reviewer 2 Report

The title of this review indicates it will cover the effects of physical activity (PA) on biomarkers of Alzheimer's disease. The content of the review in this regard is extremely superficial and not well organized.  

The introduction is appropriate and leads the reader to anticipate a thorough discussion of the effect of physical activity on AD risk.

The authors then move into a section titled AD risk factors which is very limited in this regard. Only the first two small paragraphs and the large Figure 1 of this section are are dedicated to risk factors. The majority of this section (paragraphs 3-5) describe the amyloid cascade hypothesis of AD and do so in a very superficial and limited manner. This discussion does not belong in a section titled Risk Factors as it is not a risk factor. There are several misstatements in this discussion as well, such as "abeta production and deposition by APP" (l. 69). Abeta is derived from APP, not deposited by APP. Also, "abeta 42, the most common type" (l. 72) is not correct. Abeta 40 is the most common type. 

I fail to see the relationship of Section 3 to the topic of this review as outlined in the title. It, again, is a rather superficial discussion, but if it were not included I would have not noticed it's absence. If it is left in, the authors should somehow reveal its relevance to the overall discussion of the topic. I'm not sure how this could be done.

Sections 4 and 5, the main topic of the review, I again find to be very superficial and very under referenced.  For example, a recent review of this topic by Yuede et al., (Neurobiology of Stress 8:158-171, 2018) references far more studies of PA effects on markers of AD than are covered here and there have been several more studies on this topic published since the Yuede review was published. I also found this section of the review very difficult to follow. There are many layers to the effect of physical activity on AD onset and progression. The type of physical activity matters, wheel running, treadmill running, swimming, etc.  Intensity matters, as does age at which PA begins, and how long it continues.  All these parameters are intermixed in this section in short, declarative sentences which leaves the reader confused as to the role of PA on AD.

Author Response

Please see the responses as attache file.

All the best,

Dr.Morteza Jourkesh

Round 2

Reviewer 2 Report

Moving amyloid hypothesis from the Risk Factor section makes the manuscript read significantly better.